# The Effects of Language Teaching Pedagogy on Cognitive Functioning in Older Adults

**DOI:** 10.3390/bs13030199

**Published:** 2023-02-23

**Authors:** Mara van der Ploeg, Wander Lowie, Merel Keijzer

**Affiliations:** 1Applied Linguistics Department, University of Groningen, Oude Kijk in ‘t Jatstraat 26, 9712 EK Groningen, The Netherlands; 2English Linguistics and English as a Second Language (ESL) Department, University of Groningen, Oude Kijk in ‘t Jatstraat 26, 9712 EK Groningen, The Netherlands

**Keywords:** late-life language learning, cognitive flexibility, implicit/explicit grammar instruction, language learning pedagogies, older adults

## Abstract

With the field of late-life language learning (LLLL) expanding fast, ample attention has been paid to cognitive benefits ensuing from LLLL. However, these studies have yielded mixed results, which may be partly explained by seniors’ language learning needs not being taken into account, and theoretical insights on effective language teaching have not included seniors. In order to link seniors’ language learning needs to possible cognitive benefits, and to expand the second language acquisition literature, 16 Dutch seniors took part in a three-month English course, with or without explicit grammar instruction, to ascertain the effects of more implicit versus more explicit language teaching pedagogies on cognitive flexibility. More specifically, we used linear mixed effects models to determine these methods’ differential effects on attention, working memory, processing and switching speed, inhibition, and shifting and switching abilities, as subdomains of cognitive flexibility, by using a pretest–post-test–retention test design. On the digit span tasks, the explicitly taught group showed significant improvements compared to the implicitly taught group. For Dutch verbal fluency, participants’ performance significantly improved regardless of condition. On the other measures, no differences between the groups were found. Hence, if the goal is to improve seniors’ working memory, then explicit language instruction appears more fruitful than implicit language instruction.

## 1. Introduction

The field of late-life language learning (LLLL) is expanding fast. Within this field, particular attention has been paid to cognitive benefits that may ensue from such late-age language learning. This idea was first coined by Antoniou, Gunasekera, and Wong [1], who suggested that seniors—by learning a new language—might benefit from similar cognitive advantages that have been documented for lifelong bilingualism (cf. [2] for an overview). Cognitive benefits ensuing from lifelong multilingualism have mostly been suggested to impact cognitive flexibility [3], but cognitive flexibility as a construct encompasses multiple cognitive processes: shifting, inhibition, monitoring, working memory, and attention [4]. Although generally, cognitive flexibility tends to decline as a function of ageing [5,6], it is assumed that lifelong multilinguals build up cognitive reserve due to an improvement in cognitive control (cf. [7] for an elaborate discussion of the construct), which may delay the symptom onset of degenerative diseases, such as Alzheimer’s [8].

LLLL work has attempted to ascertain if cognitive benefits associated with late-life language learning can resemble those found in lifelong multilingualism. However, the studies conducted so far have yielded mixed results (cf. [9,10,11,12,13,14,15,16]; for an overview of the studies, see [17]), which may be partly explained by the fact that seniors’ language learning needs have not been structurally taken into account [17]. Indeed, as a relatively new learner demographic, insights obtained from applied linguistic investigations into effective foreign language pedagogies have been predominantly based on younger learners. Until recently, seniors’ language learning needs have largely remained unclear (cf. [18]), but may well be radically different from those characterising younger language learners. In a recent study by van der Ploeg and colleagues [19], 197 seniors filled out a questionnaire indicating their language learning preferences. The results showed that, amongst other things, seniors prefer any late-life language course to comprise speaking exercises, error corrections, grammar explanation, and to build on a teacher-focused teaching style. Even though these needs relate to teaching pedagogies, past LLLL studies have, however, not included second-language acquisition (SLA) theories, such as the widely-researched theoretical constructs of implicit versus explicit teaching methods (see below; [17]). In short, although a focus on cognitive outcomes of late-life language learning may be relevant, to move the field forward a more fundamental analysis of language pedagogy is needed, in which the effectiveness of methods, as well as learner preferences, are directly related to cognitive outcomes of LLLL incentives.

### 1.1. Applied Linguistic Approaches to Implicit and Explicit Instruction

The difference between more implicit and more explicit language instruction is best characterised in terms of a difference in the metalinguistics input (“learning (about) the language”) learners do or do not receive [20,21]. This can be further broken down into the much researched distinction between *focus on form*, *focus on forms*, and *focus on meaning* [22]. *Focus on form* has been coined as the most prevalently used teaching method, that typically prioritises conveying a message, but is also a method in which the teacher pays explicit attention to the form of the message in the process. In contrast, *focus on forms* can be characterised as a traditional, explicit grammar/translation teaching method, largely with an absence of meaningful interaction. *Focus on meaning*, then, tends to focus exclusively on communicative competence [23], with an absence of any attention to form. Although focus on meaning tends to be equated with implicit language instruction, where the learner is expected to learn as a result of natural and abundant (often repeated) exposure and input, it is important to note that implicit versus explicit language instruction is a continuum and, related to that, researchers have claimed that fully implicit or explicit instruction does not exist [24,25].

Many studies investigating the effectiveness of implicit versus explicit methods have been conducted, but a consensus as to the most optimal/effective method is hard to reach [26]. Some of the earlier studies showed a clear advantage of explicit language instruction, as summarised in the metastudies by Norris and Ortega [21] and Spada and Tomita [27]. However, these earlier studies have been criticised because: (1) they targeted one specific grammatical structure [28], (2) the training period was short, and, (3) the task procedures and test circumstances tended to favour explicit knowledge [26]. It needs to be pointed out that the work on implicit versus explicit teaching methods to date has been based on younger learners and cannot be directly translated to senior learners. In addition, earlier work has not directly related teaching methods to cognitive outcomes, as this is less pertinent for younger adults. In the present study, we uniquely relate cognitive outcomes of late-life language learning to the implicit versus explicit teaching instruction continuum. This is all the more important as there are interesting parallels to be drawn here to implicit and explicit memory, with memory known to decline in ageing.

### 1.2. Psychological Approaches to Implicit and Explicit Learning and Memory

Building on the work investigating the effectiveness of implicit versus explicit language teaching approaches, “cognitive psychology suggests that types of L2 (second language) instruction with higher cognitive demands may impair older adult learning” [29] (p. 30). This is mainly related to the reduced working memory (WM) capacity [30] and reduced inhibitory control [29], that generally accompany ageing. However, these cognitive skills are needed in noticing and pattern identification, both of which are important in learning a new language, and are especially pertinent in implicit approaches to language learning [31]. In the absence of empirical evidence, it can be assumed that implicit language instruction might not be as effective in older adults, compared to more explicit instruction methods.

Beyond the confines of language instruction, explicit and implicit learning more generally refer to intentional versus unintentional learning, respectively, and how both of these constructs relate to ageing has been widely investigated. Domain-general explicit learning ability has been found to decline with age [32], whereas this decline has not consistently been found for implicit learning e.g., [33,34,35].

Implicit and explicit learning need to be separated from implicit and explicit knowledge, which can be described as doing versus knowing [36]: being able to apply knowledge versus verbalising it. Research has found explicit knowledge and memory to decrease with age, whereas implicit knowledge and memory in older adults are generally intact, and comparable to younger people [37,38,39]. In a study by Rebok, Rasmusson, and Brandt [40], twelve older adults (*M_age_* = 76.33) were provided with either implicit or explicit computerised memory training (Colorado Neuropsychology Tests v2.0 software; [41]) for 1.5 h each week, for nine weeks. Although participants’ memory in both conditions improved, as measured by means of the Colorado Neuropsychology Tests, the implicit group improved most. As an explanation, the authors stated that the implicit memory training’s effectiveness “may be due to the fact that they involved learning new skills, including using the computer mouse” (p. 218) although the explicit condition also completed a computerised training. Similar results were found by Vakil, Hoffman, and Myzliek [42]. In their study, 24 older adults received either ‘active’ (standard administration) or ‘passive’ (instructions read to them) training on the Tower of Hanoi task, a procedural learning task. Results showed that older participants did better on the active training task, which is more closely related to implicit learning, as no rules/instructions were provided. It does, however, need to be noted that the Tower of Hanoi task is considered to be an implicit memory task, and this might therefore have biased the results towards the ‘active condition’.

Even though the domain-general learning literature suggests that implicit learning might be more beneficial for older adults, we also know that language learning is substantially different from other types of learning. What is more, results from cognitive psychology and applied linguistics at first may seem contradictory, with the former pointing to implicit instruction as benefiting cognitive outcomes [40,42] and the latter pointing to explicit instruction benefiting language development in older adulthood because of compromised working memory capacity (cf. [29,30,31]). It needs to be emphasised, however, that studies into additional language learning have not explicitly targeted cognitive outcomes as a function of the implicit–explicit instruction continuum. Hence, the present study aims to investigate implicit and explicit language instruction studies in older adults, and specifically in relation to cognitive outcomes of late-life language learning.

### 1.3. Implicit and Explicit Language Instruction Studies in Older Adults

Several studies to date have investigated the effect of language teaching pedagogies on older adulthood, be it with a focus on language, socio-affective, or cognitive outcomes (for an overview see [43]). Of the 20 LLLL studies (we are aware of) that have been carried out to date, very few studies have investigated implicit and explicit language instruction in seniors [43].

The work by Cox and Sanz [29] and Cox [44] solely focused on explicit language instruction in basic Latin morphosyntax lessons. Cox and Sanz [29] found that younger bilinguals (19–27) benefited more from explicit instruction than older bilinguals (60+), whereas Cox [44] found no overall effects of explicit instruction, as compared to only practice, in older monolinguals and bilinguals. These results seem to indicate that explicit grammar instruction might not be the most effective method for older adults. This is confirmed by two studies that included both types of language instruction, the works by Midford and Kirsner [45] and Lenet, Sanz, Lado, Howard Jr., and Howard [46]. Although the methodologies used in these studies differed, both found seniors to perform better in a less explicit condition.

The study by Midford and Kirsner [45] compared performance on an artificial grammar learning task, comparing younger (17–28 years) and older adults (58–74 years). This task comprised two conditions: an explicit condition, where the grammar was schematically presented, and an implicit condition, without any such overt instruction and merely exposure to the artificial language. Participants were tested on recognition accuracy of correct and incorrect grammar rules, and the authors found the older group to do better in the implicit condition than their younger counterparts. Lenet and colleagues [46] carried out a similar experiment, but in their study, seniors (66–81) received Latin vocabulary training followed by grammar training that incorporated either implicit or explicit feedback; the implicit feedback, comprising correct/incorrect-only feedback, was more effective than the explicit feedback, which included elaborate grammar explanations. It needs to be noted that what was termed implicit feedback here was still rather explicit. Recasts, for example, would have been much more implicit.

### 1.4. The Present Study

Although there is evidence that implicit and explicit learning and memory are differentially affected by ageing in a general sense, the limited existing work specifically targeting implicit versus explicit language instruction, and their differential effects on cognitive improvement or maintenance, has produced mixed findings. Therefore, in this small-scale longitudinal study, we investigate the effect of implicit and explicit grammar instruction on cognitive functioning in older adults as a way of substantiating the field. Our research question was: do largely implicit and largely explicit language instruction methods in older adulthood result in different cognitive flexibility outcomes? Even though the literature described above points to older adults doing better on implicit (language) learning, when focusing on the cognitive domain, we also know that WM and inhibition are generally compromised in older adults (cf. [29,30]). With mixed findings being what they are, this study is exploratory in nature, without postulating specific hypotheses. Indeed, we might well hypothesise either implicit or explicit grammar instruction to have a bigger effect on cognitive flexibility at an advanced age: while training studies have shown implicitly trained seniors to improve most in memory [40], reduced WM and inhibition skills are found in older adults (cf. [29,30]), yet are important in implicit language learning [31]. Furthermore, implicit language teaching methods have been found to be more effective for seniors when it comes to proficiency (cf. [45,46]).

## 2. Materials and Methods

### 2.1. Participants

Sixteen Dutch seniors participated in our longitudinal study (*M_age_* = 71;9, *SD_age_* = 6;3, 13 female, 3 male). Their educational levels varied from having completed secondary education to university education, and their former professions included police officer, secretary, teacher, and business owner. All participants were retired at the start of the study. They were all taught English as part of the language learning intervention. A total of 30 seniors started the study, during the COVID-19 pandemic. However, 14 of them dropped out of the study due to: health issues (*N* = 6), difficulty with online technology (*N* = 4), experienced course difficulty (*N* = 3), and a strong dislike of the assigned teaching pedagogy (implicit; *N* = 1). Participants were recruited through senior organisations in the Netherlands, and Facebook advertisements. Inclusion criteria were: (a) aged 65–80, (b) a cutoff at B1 level (Common European Framework of Reference for Languages; CEFR) for English proficiency, and (c) no known cognitive problems (based on a score of 26 or higher on the Montreal Cognitive Assessment (MoCA); [47]). The study was approved by the Research Ethics Committee at the University of Groningen (61890455).

### 2.2. Design and Procedure

All seniors were enrolled in a three-month (following [1]) online English course, specifically targeted at seniors, with materials created specifically for this course. The online learning environment was necessitated by the timing of this study, during the COVID-19 pandemic lockdown. Supported by the findings of Kueider, Parisi, Gross, and Rebok [48], we adapted the online learning environment to suit our seniors’ needs, which included adding explanations regarding technology, using low-threshold technology, and providing space for social interaction [49].

Participants were randomly assigned to one of two conditions: a mostly implicit or a mostly explicit English course condition. ‘Purely’ explicit versus implicit language teaching is impossible [24,25]. Our explicit condition was not fully explicit, in the sense that only grammar was explained in the absence of other language input. Similarly, the implicit condition was not fully implicit, as explicit attention was sometimes given to vocabulary items, and the L1 Dutch translation was provided if participants asked for it. Our methods are perhaps best characterised as more form-focused (explicit condition) rather than more meaning-focused (implicit condition). Attention to meaning and communicative competence is much more present in the mostly implicit condition [22].

Specifically, we offered two versions of the same English course, one where the (online) classroom materials and homework included explicit explanations and mentions of grammatical constructs, and one where these explanations and mentions were replaced with extra input of the target structure. For example, in the explicit course the present simple tense was explained as: “we use the present simple to talk about things that happen regularly in the present”, followed by a breakdown of the sentence structure. In the implicit course we provided several example sentences describing habits, as part of a communicative exercise. An excerpt from both versions of the course can be found in Figure 1.

In both conditions, participants participated in weekly two-hour Zoom group lessons, and they completed approximately 60 min of homework assignments (following [9]) for five days each week. The homework consisted of watching videos and answering questions about the videos, learning new vocabulary, and speaking and writing about different topics including hobbies, climate change, and public transport. During the lessons, most of the time was spent on speaking, as seniors had explicitly indicated this preference as part of a precursor study [19]. In addition to the homework assignments, participants completed daily diary entries, resulting in dense motivation, well-being, perceived learning amount, and extracurricular English activities data.

Prior to course onset, participants were tested using a combination of cognitive (Table 1; in their L1) and language measures (i.e., listening, speaking, vocabulary, and verbal fluency). Additionally, they completed several questionnaires tapping into their socio-affective states (i.e., motivation and well-being; in their L1). This same test battery and procedure was repeated immediately after the course (post-test), and as part of a retention test three months after the post-test. In this paper we solely focus on the cognitive outcomes; the language and socio-affective outcomes are reported in upcoming papers.

The cognitive tests were based on an earlier LLLL protocol by Nijmeijer and colleagues [51], and are described in more detail in Appendix A, but include attention and associative learning, WM, switching, shifting, and inhibition. Testing was completed using Google Meet by the first author. Physical copies of the paper–pencil tests (MoCA, TMT, and DSST) were sent to participants’ homes in a closed envelope, with a note specifying that the envelopes should only be opened when participants were instructed to do so by the researcher. All participants complied with this request. After completing the test, participants were asked to hold up the piece of paper in front of the webcam in order for the researcher to screenshot the data for scoring and, next, throw out the paper. The CSST and mWCST were administered in OpenSesame, version 3.1.9 [52]. For the CSST, four different versions were used in a counterbalanced manner, where the order of task blocks, and the hand participants used to respond to the stimuli, were varied.

The experiment (comprising the test battery and English course) was piloted in two rounds, in which two and eight seniors, respectively, participated (respectively *M_age_* = 77;9, *SD_age_* = 3;6, 2 male; and *M_age_* = 70;9, *SD_age_* = 4;3, 2 female, 6 male), none of whom participated in the actual study. Although the setup of the course proved to be mostly adequate, some minor alterations were made, such as fewer homework assignments and a different homework platform.

### 2.3. Analysis

To analyse the data we ran linear mixed model analyses in R, using the lme4 package [53]. The different cognitive test scores were included as the dependent variables, and test moment (pre-, post-, and retention test) and condition (explicit and implicit) were included as independent variables, with an interaction. Finally, participant was included as a random effect, as older adults are known to show variation (cf. [54,55]). Using the mgcv package [56], the conditional R^2^ was calculated. The scores for all cognitive tests were centred around zero based on the pretest scores. Graphs were created using ggplot2 [57]. See Appendix B for model output from all models. Finally, the data from the CSST and the mWCST were not analysed. We opted out of analyses for these tasks due to a combination of large variability (see Results) in participants’ test scores, and substantial missing data, leading to model convergence issues: 37 data points for the mWCST instead of 48, and 26 data points for the CSST instead of 48 (number of missing data in pretest: two mWCST, seven CSST; missing data in post-test: two mWCST, five CSST; missing data in retention test: seven mWCST, ten CSST). The reasons for the missing data were all related to participants’ inability to complete the tasks, and included physical problems such as rheumatics, and technological shortcomings such as use of tablets and not being able to open the experiment (despite having piloted the online tasks prior to the study).

## 3. Results

To test the effect of language teaching pedagogy on cognitive flexibility, linear mixed models were run for each cognitive measure. The results section describes the outcomes of these measures per cognitive flexibility construct.

### 3.1. Working Memory

In the digit span forward task, we found a significant effect between time and condition and digit span forward score (conditional R^2^ = .61; Table A1). Our model showed a significant interaction between both post-test and retention test and condition (*b* = −1.87, *SE* = 0.80, *t* = −2.34, *p* < .05, CI[−3.45, −0.27]; *b* = −2.50, *SE* = 0.80, *t* = −3.12, *p* < .01, CI[−4.00, −0.93]), meaning that the explicit condition significantly outperformed the implicit condition on the post-test and retention test, compared to their pretest scores (*b* = −1.87, *SE* = 0.83, *t* = −2.25, *p* < .05, CI[−3.66, −0.08]; *b* = −2.50, *SE* = 0.65, *t* = −3.87, *p* < .01, CI[−3.88, −1.11]). However, no significant differences were found between groups between post-test and retention test (*p* = .44). In other words, the comparatively higher score for the explicit group was maintained, but did not increase further. Figure 2 depicts the development of the scores over time per condition. The substantial error bars seen in Figure 2 can be ascribed to the individual differences that were attested. In all models described in this section we found big individual differences, meaning that all error bars presented are substantial.

Similar to the digit span forward outcome, the digit span backward sequences also increase in length. Again, we found a significant effect between time and condition and digit span backwards (conditional R^2^ = .56; Table A2). Similar to the digit span forward, we found a significant interaction between retention test and condition (*b* = −2.13, *SE* = 0.95, *t* = −2.23, *p* < .05, CI[−4.26, −0.03]), meaning that the explicit condition performed significantly better than the implicit condition on the retention test (*b* = −2.13, *SE* = 0.88, *t* = −2.42, *p* < .05, CI[−1.01, −0.24]). In this model, for the post-test, a trend towards a significant interaction with condition was found (*b* = −1.75, *SE* = 0.95, *t* = −1.84, *p =* .07), indicating that the explicit condition did better on the post-test, but this difference only reached significance at the retention test. No significant differences were found between groups between post-test and retention test (*p* = .69). Figure 3, below, visualises the development per condition over time.

Finally, in the digit span letter number sequencing, again, the sequences increased in length over time. We found a significant effect between time and condition and digit span letter number sequencing (conditional R^2^ = .37; Table A3). We found a significant interaction between both post-test and retention test, and condition (*b* = −2.38, *SE* = 1.13, *t* = −2.10, *p* < .05, CI[−4.48,−0.06]; *b* = −2.38, *SE* = 1.13, *t* = −2.54, *p* = .01, CI[−4.66, −0.33]), indicative of the explicit condition performing significantly better than the implicit condition on both the post-test and the retention test (*b* = −2.38, *SE* = 0.79, *t* = −3.02, *p* < .01, CI[−4.06,−0.69]; *b* = −2.89, *SE* = 0.59, *t* = −4.87, *p* < .001, CI[−4.14, −1.61]). No significant differences were attested between groups between post-test and retention test (*p* = .66). Figure 4, below, visualises the development per condition over time.

### 3.2. Inhibition

For Dutch verbal fluency, we found a significant effect between time and verbal fluency scores (conditional R^2^ = 0.76; Table A4). Our model showed a significant main effect for both post-test and retention test (*b* = 6, *SE* = 2.24, *t* = 2.48, *p* < 0.05, CI[0.94, 10.24]; *b* = 7, *SE* = 2.24, *t* = 2.89, *p* < 0.01, CI[1.70, 11.94]), which means that, regardless of condition, older adults recalled more items as part of the Dutch verbal fluency test after their language course. No significant differences were found between groups between post-test and retention test (*p* = 0.68). See Figure 5, below, for a visualisation of the model.

### 3.3. Other Measures

For the digit symbol substitution task and the trail making test, no significant differences of either time, condition, or an interaction between the two were found in our models. Figure 6, Figure 7 and Figure 8, below, visualise the development per condition over time.

## 4. Discussion

In this small-scale longitudinal study, we set out to investigate whether seniors learning English as part of a more explicit versus implicit condition differ in cognitive flexibility that may ensue from such a training. In the emerging field of third-age language learning (LLLL), ample attention has been paid to cognitive benefits that may ensue from LLLL, but this has not rendered uniform results. Due to the mixed findings, we did not postulate specific hypotheses, and instead adopted an explorative design as one of the first studies to directly compare a more implicit or more explicit teaching method. In our approach, we combined insights from applied linguistics and the psychology of learning and memory, and applied these to an older adult life stage. Our results show that in the digit span tasks the explicitly taught group outperformed the implicitly taught group. As the digit span task is an index of WM, we might conclude that, if the goal is to improve WM as a component of cognitive flexibility, explicit grammar instruction might be more effective. Looking at the literature, the reduced WM capacity that is generally observed in seniors [30] might help explain our findings of WM being important in implicit approaches to language learning [31]. Yet, although inhibitory control is also known to decrease with age [29], no differences between our conditions were found on our Dutch verbal fluency task. We did, however, find an effect of time: older adults, in both the conditions of our study, obtained better results for the Dutch verbal fluency scores over time. This might mean that any kind of language learning can help boost inhibitory skills. Nonetheless, except for the digit spans, none of our tasks showed any significant difference between the groups (note that we did not analyse the mWCST and the CSST due to missing data). Of course this does not mean that such an effect does not exist in a study that has more power to find it. One possible explanation for the lack of findings in most tests is the substantial individual variation, which is generally accepted in gerontology; as put by Grotek [55], “this age group is characterised by the largest diversity of any age groups involved in education” (p. 128). Moreover, inter- and intra-individual differences in cognition are known to increase over the lifespan [54], and cognition and L2 performance can fluctuate within an individual on a day-to-day basis (cf. [54,58,59]]. This makes it hard to generalise across these participants and to speak of “the late-life language learner”. However, the individual variance does not provide an answer to the question of why differences were found for WM but not for other cognitive domains. It might be that explicit language instruction has a particular effect on WM. Indeed, many LLLL studies have found a relationship between language learning and WM (cf. [60,61]). Yet, Berggren and colleagues [16] found no effect of LLLL on WM, and concluded that “basic studies in foreign languages in older age are likely to have no or trivially small effects on cognitive abilities” (p. 212). This, then, amplifies the need to uncover seniors’ language learning needs, as attempted in this study, to move away from these ‘basic studies in foreign languages’, as we have shown different language teaching pedagogies to have different effects on WM.

### 4.1. Limitations

The findings of our study need to be interpreted with care, due to a number of choices related to scope, and practical limitations. For one, our sample was relatively small, and even though there were good reasons for this—COVID-19, the study having to take place online, and it being a longitudinal study—it does mean that our results need to be interpreted with caution, and that future research should replicate our findings with larger sample sizes. The fact that 14 participants dropped out of the study does, however, show that language learning takes effort, which is of course needed in order for potential effects of language learning to become visible [1,9].

Moreover, the online nature of the study is bound to have impacted the results, due to potential issues with internet connections, technical difficulties, and auditory difficulties. Although the majority of testing sessions and language lessons went as smoothly as possible, given the circumstances, the two computer tasks (mWCST and CSST) suffered from the fact that no researcher was physically present to help the participants. Finally, the online nature of the study also led to a bias in participant selection, as people needed to have the digital skills to join the study, or at least believe they had these skills. A final limitation we want to touch upon is the duration of the language course. It has been noted that finding effects of implicit language pedagogies on proficiency may take a longer time, as more input is needed [62]. It might be expected then that the cognitive effects of implicit language instruction pedagogies also take longer. Hence, we suggest future studies to prolong the language training period to test this hypothesis. Moreover, implicit language development is known to demonstrate a U-shaped development; before language proficiency improves, it gets worse (cf. [63]). This, then, might be the same for cognitive development resulting from implicit grammar instruction.

### 4.2. Scientific and Practical Implications

Finally, we want to point to some scientific and practical implications of this study. First of all, our study shows explicit instruction to have a bigger impact on WM than implicit instruction, leading us to suggest explicit grammar instruction if improvements in cognitive flexibility are the goal. Turning to inhibition skills, however, we did not find the type of language instruction to matter, and both groups improved over time. This leads us to our second point: individual variation. Individual differences are known to be important in younger learners (cf. [64]), yet we might argue they are even more important in older adults (cf. [54]). Indeed, we observed individual variation in our study that is in line with this literature. Therefore, future research should take this into account by not only treating seniors as a heterogeneous group, but also by investigating possible factors contributing to individual differences, to create a more complete picture of the late-life language learner and their cognitive functions. This also holds true for LLLL teachers; seniors should not be treated as a homogenous group but attention should be paid to the individual. This is further corroborated by classroom studies showing seniors’ agency over their language learning process [65]. Next, we believe this study has shown that it is possible to teach and study seniors using online tools. However, the CSST was reported to be unpleasant by multiple participants—not only physically unpleasant, because of rheumatics, but also mentally, because of duration and intensity—and we would, therefore, not recommend using it in future studies. Additionally, these computerised tasks might work well with seniors coming into the lab, or when a researcher is present, but might not work equally well when they have to be completed by seniors without any help. Troubleshooting remotely is difficult and many seniors might not have a PC/laptop but only a tablet. Many computerised tasks, however, are not (yet) compatible with tablets.

## Figures and Tables

**Figure 1 behavsci-13-00199-f001:**
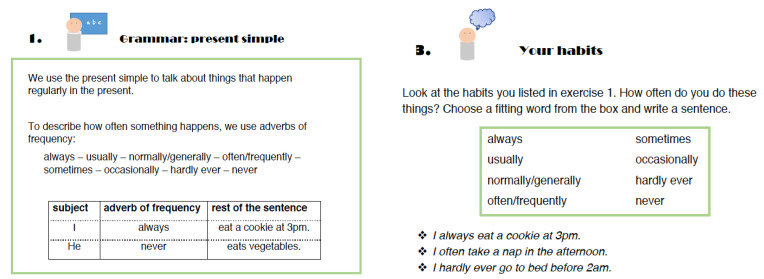
Explicit (**left**) and implicit (**right**) explanation of present simple.

**Figure 2 behavsci-13-00199-f002:**
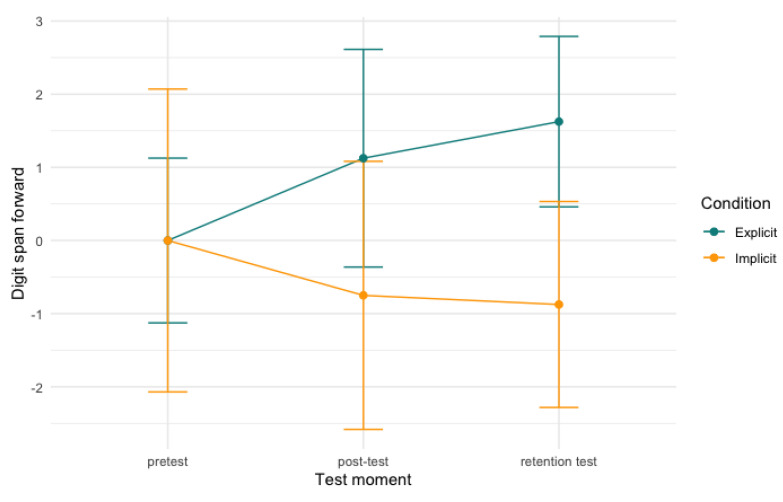
Digit span forward scores per condition over time. Note: scores are centred around zero.

**Figure 3 behavsci-13-00199-f003:**
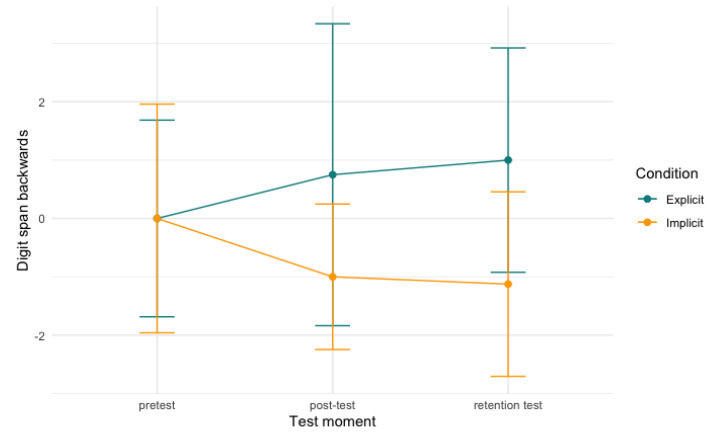
Digit span backwards scores per condition over time. Note: scores are centred around zero.

**Figure 4 behavsci-13-00199-f004:**
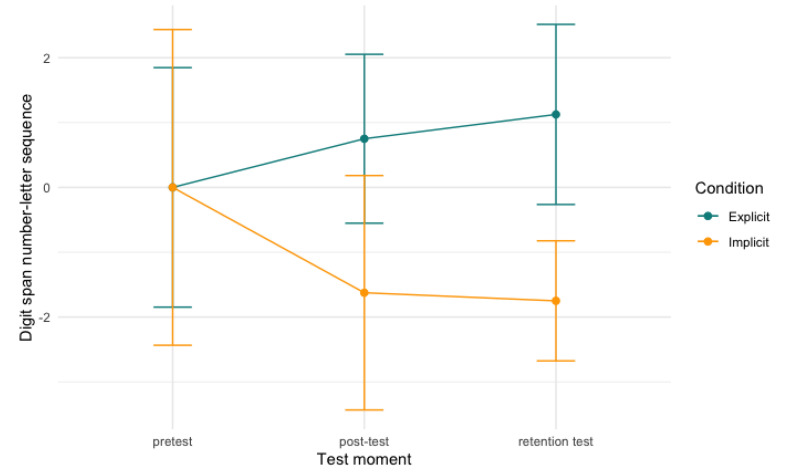
Digit span letter number sequencing scores per condition over time. Note: scores are centred around zero.

**Figure 5 behavsci-13-00199-f005:**
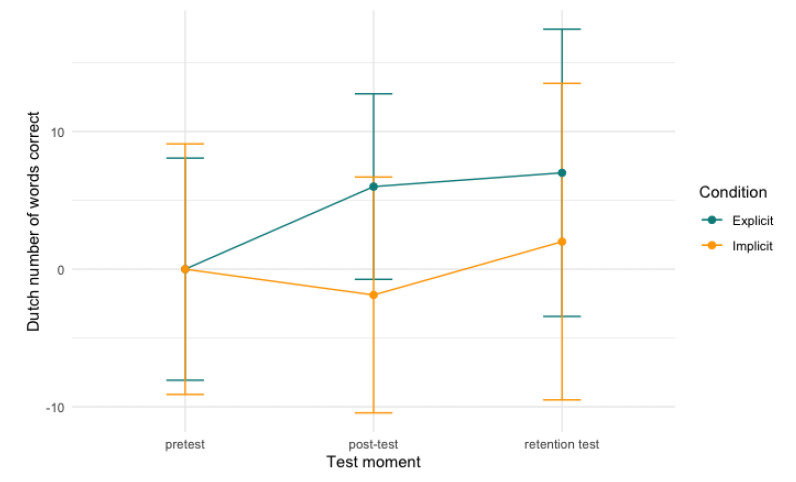
Dutch verbal fluency scores per condition over time. Note: scores are centred around zero.

**Figure 6 behavsci-13-00199-f006:**
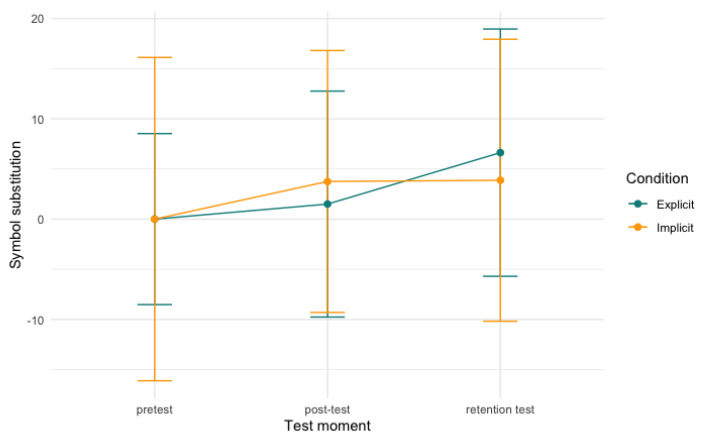
Symbol substitution scores per condition over time. Note: scores are centred around zero and the model is not significant.

**Figure 7 behavsci-13-00199-f007:**
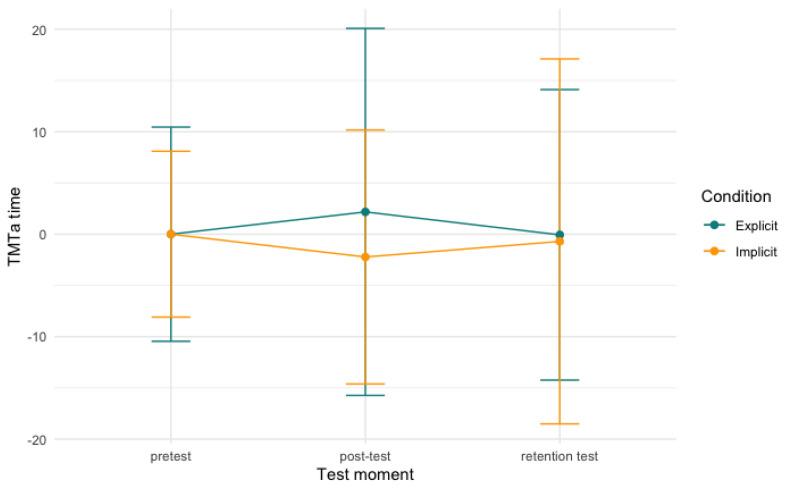
Trail making test part A time per condition over time. Note: scores are centred around zero and the model is not significant.

**Figure 8 behavsci-13-00199-f008:**
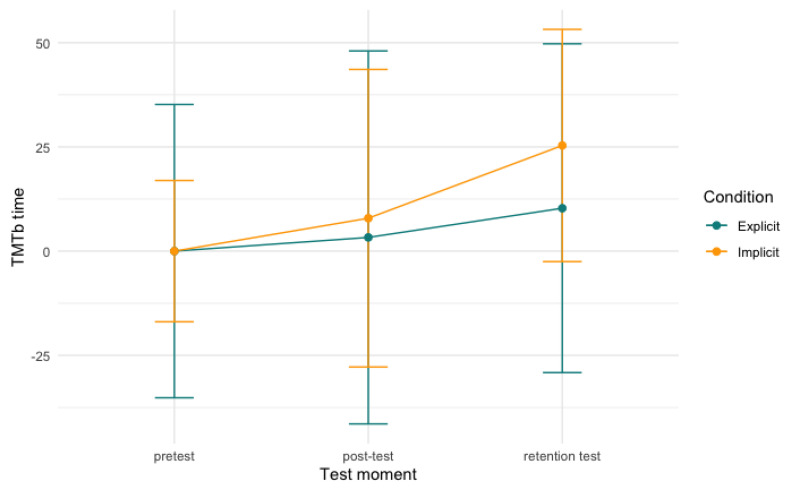
Trail making test part B time per condition over time. Note: scores are centred around zero and the model is not significant.

**Table 1 behavsci-13-00199-t001:** Overview of cognitive measures and their respective cognitive flexibility construct.

Cognitive Measures	Cognitive Flexibility Construct
MoCA [47]	cognitive screening test for cognitive functioning
Trail Making Test (TMT) parts A and B	processing and switching speed
Modified Wisconsin Card Sorting Test (mWCST)	shifting abilities
Colour Shape Switching Test (CSST)	switching abilities
Dutch verbal fluency	executive control/inhibition
Digit symbol substitution (DSST) (WAIS-IV; [50])	attention and associative learning
Digit span tests (WAIS-IV; [50])	working memory

## Data Availability

The data presented in this study are available on request from the corresponding author.

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
