# Peer review of "The Effects of Language Teaching Pedagogy on Cognitive Functioning in Older Adults"

_behavsci, 2023, doi:10.3390/bs13030199_

Round 1

Reviewer 1 Report

The study will be of interest to educators and researchers as it deals with the issue of language learning in terms of explicit vs implicit teaching for the aging population for developed countries – an under explored area. The paper has a well written introduction to contextualize the research and relevant literature review. There is detailed description of methodology and clear reporting of findings.

However, as a statistical study, there exists the issue of deriving statistical significance for only 16 participants as acknowledged by the authors as a limitation of the study.

Reviewer 2 Report

The authors report on a longitudinal online study that examined whether implicit versus explicit language instruction differentially impacted executive function in 16 older adults. They found that participants who received explicit language instruction performed better on the digit-span task at post-test and retention compared to pre-test. Those who received implicit instruction did not show this improvement. The authors concluded that explicit language instruction in older adults may confer a benefit to working memory in older age.

The manuscript is well-written and the results presented very clearly. I have one overarching suggestion plus a few minor considerations for the authors.

General Comments

-I understand that the aim of the study was to examine how language learning impacted cognition beyond language, but it seems useful to include language proficiency outcomes. It would be relevant to know whether implicit/explicit instruction impacted language learning after three months, for example, and whether this was related to cognitive differences (i.e., on the digit span task).  

Specific Comments

-What were the socio-affective measures? These are not listed in the Table 1 or appendix and aren’t discussed.

-Analysis section: it would be helpful to state the aim of the analyses here (i.e., “To test the effect of learning condition on each cognitive measure, linear mixed models were run…”)

-Can you confirm that the cognitive tests were in L1? Please specify in the methods.

-Was any test of language proficiency administered? I understand the aim to look at how implicit/explicit instruction affects other cognition, but which was more effective for language learning (if any)? As mentioned above, it would be useful to know if the cognitive differences emerged in relation to or in the absence of any language learning differences.

-Working memory results section: the wording is a bit confusing when describing the interactions. With how they are written now, it is confusing to say, “…meaning that the explicit condition significantly outperformed the implicit condition on the post-test and retention test compared to their pretest scores” then mention “No significant differences were attested between groups between post-test and retention test.” I’m not quite sure what these sentences mean. Was the significant difference between timepoints within explicit participants only? Consider rephrasing. It would also be helpful to depict the significant differences in the figures.

-p9 l343 Inhibition section: there seems to be a typo at the start of this section, which includes a Figure 2 caption.

-Discussion: why would the goal of language instruction be to improve WM? Perhaps it’s a nice side effect of explicit instruction, but without knowing the impact on language learning, it’s hard to deduce whether this is useful or not. If WM is related to language learning, that’s even more reason to include a language learning outcome and look at relationship.

Reviewer 3 Report

The authors developed this study with the aim of linking the language learning needs of older adults to possible cognitive benefits. To do this, they first conducted a presentation of the field of third-age language learning (TALL), with its linguistic approaches to implicit and explicit instruction. Then they used their study to determine the differential effects of these methods on attention, WM, speed of processing and switching, inhibition, shifting ability, and switching as subdomains of cognitive flexibility, using a pre-test/post-test/retention-test design. This is an interesting idea that can increase the knowledge of methodologies that can help maintain and improve cognitive skills in elderly subjects.

The organization of the paragraphs is clear and smooth. They are detailed, well-constructed and perfectly focuses on the theme of the paper with adequate bibliographic references. The data are well-presented and represented through the figures.

However, I ask the authors to consider the following comments to improve and implement the quality and robustness of the manuscript.

Comment 1: The quality of English needs to be improved. Sometimes the complexity in the construction of the sentences makes reading not very smooth and fluid. Authors are encouraged to do a minor review of the manuscript. The use of appropriate terminology is critical to the context. Authors are encouraged to rephrase some sentences and review the use of some terms.

Comment 2: The reading of the text is not clear and fluid at times, due to the lack of explanation of the acronyms. Here are some references: in the Abstract the acronyms SLA and WA; line 103, acronyms L2; line 209 acronym CEFR; line 211 acronym MoCA.

Comment 3: Authors are asked to pay attention to citations in the text. There are several statements in the text that are not supported by bibliographic references. The authors are invited to modify the content, inserting relevant and adequate references. Here are the lines interested: line 56, “In a recent study by Authors (submitted)”, please explain the name of the authors and when and where it was submitted; line 139-142 “What is more, results from cognitive psychology and applied linguistics at first may seem contradictory, with the former pointing at implicit instruction as benefiting cognitive outcomes and the latter pointing at implicit  instruction benefiting language in older adulthood because of compromised working memory”, please insert the citation that allows you to link your statement to the literature; line 188-189, “we also know that WM and inhibition are generally 188 compromised in older adults”, please insert the citation that allows you to link your statement to the literature.

Comment 4: Some sections of the text are not elaborated or clearly expressed to be immediately understood. Here are some tips to make the text and therefore the search clearer. In line 152-153, authors affirm “Of the 20 TALL studies (we are aware of) that 152 have been carried out to date”; please indicate the bibliographic reference where you found this data, or explain how you are able to quantify the works existing in the literature on this topic in this number. In line 200, “(M = 71;9, SD = 6;3, 13 female)”, the meaning of the text inside the brackets is not entirely clear; Please explain more clearly the number of men and the number of women who participated in the study. Line 235-238, “In line with what has been pointed out before, ‘purely’ explicit vs. implicit language 235 teaching is impossible. Our explicit condition was not fully explicit in the sense that only 236 grammar was explained in the absence of other language input. Similarly, the implicit 237 condition was not fully implicit”; In order for the reader to understand the objective and the methodologies used in your study, there must be clarity in the distinction between the explicit method and the implicit method. It is therefore necessary to explain in detail which were the explicit and implicit teaching methods, although it seems difficult to make a clear distinction between the two methodologies. If this is not clearly explained, then it is not even possible to adequately understand the meaning of the statistical results and to understand their significance, especially for the effects on cognitive functions.

Line 252, the authors state that the administration of psychometric instruments occurred. The authors are requested to give an explicit answer in the text to the following questions: who was involved in the administration in this phase? How were the tools administered and where? If the administration took place online, since some paper-pencil test protocols are present, were the standardized versions for remote administration used? If yes, please insert the bibliographic references of the test versions used. In line 283, the authors say some data was not processed because study participants did not complete the trials. At what stage of the research were the trials not completed? Please elaborate on this aspect.

Comment 5: Figure 1 is not clearly visible and it is not possible to understand either the graphic content or the textual content. In line 343, there is an error in the numbering of the figure referred to in the paragraph. The first 5 citations in the bibliography do not have the names of the authors or the years of publication. Please correct the statement.
